# Feasibility and Reliability of Quadriceps Muscle Power and Optimal Movement Velocity Measurements in Different Populations of Subjects

**DOI:** 10.3390/biology13030140

**Published:** 2024-02-22

**Authors:** Tomasz Kostka, Joanna Kostka

**Affiliations:** 1Department of Geriatrics, Medical University of Lodz, Plac Hallera 1, 90-647 Łódź, Poland; 2Department of Gerontology, Medical University of Lodz, Milionowa 14, 93-113 Łódź, Poland; joanna.kostka@umed.lodz.pl

**Keywords:** Sarcopenia, older adults, patients, aging, force-velocity, optimal shortening velocity

## Abstract

**Simple Summary:**

The feasibility and reliability of measurements of maximal short-term power (P_max_) and corresponding optimal movement velocity (υ_opt_) with a friction-loaded cycle ergometer have not been systematically assessed in older subjects and those with diseases. In the present study, all the tests of relative and absolute reliability indicated very good repeatability and low indices of error. Our results show that a friction-loaded cycle ergometer instrumented with a strain gauge and an incremental encoder may be an excellent candidate for future clinical studies in older subjects and those with diseases.

**Abstract:**

This study aimed to assess the feasibility and reliability of quadriceps maximal short-term power (P_max_) and corresponding optimal movement velocity (υ_opt_—velocity at which the power reaches a maximum value) measurements in different populations of subjects. Five groups of subjects, fifty participants in each group, took part in the study: students; patients of the cardiac rehabilitation program; patients after stroke; older adults; and subjects of different ages who performed repetitive measurements with two different bicycles. The correlations calculated for the pairs of scores ranged from 0.93 to 0.99 for P_max_ and from 0.86 to 0.96 for υ_opt_ (all with *p* < 0.001). Intraclass Correlations Coefficients (ICCs) varied from 0.93 to 0.98 for P_max_ and from 0.86 to 0.95 for υ_opt_. The standard error of measurement (SEM) varied from 16.9 to 21.4 W for P_max_ and from 2.91 to 5.54 rotations(rot)/min for υ_opt_. The coefficients of variation (CVs or SEM%) for P_max_ and υ_opt_ in the stroke group were 10.6% and 11.4%, respectively; all other CVs were clearly lower than 10%. The minimal detectable change (MDC) varied from 46.6 to 59.3 W for P_max_ and from 8.07 to 15.4 rot/min for υ_opt_. MDC% varied from 9.53% to 29.3% for P_max_ and from 8.19% to 31.7% for υ_opt_, and was the highest in the stroke group. Therefore, the precision of measurements of P_max_ and υ_opt_ was confirmed by very good indices of absolute and relative reliability. The proposed methodology is precise, safe, not time-consuming and feasible in older subjects and those with diseases.

## 1. Introduction

Age-associated muscle mass and strength loss is one of the most important factors connected with the functional status of older people [1,2]. Even more important than muscle strength factors affecting functional performance in older and diseased populations are muscle power and muscle contraction velocity [3,4]. These factors are also more influenced by aging than muscle strength [5]. In particular, lower extremity function and leg extensor power have been connected with physical functioning and age-related impact in older adults [6]. Measurements of force–velocity profiles and muscle power are less frequently performed than those of muscle strength or functional tests that involve both strength and power contributors. This is because more sophisticated and expensive equipment is usually needed. 

One of the possible methods of lower-extremity power measurement is with a friction-loaded cycle ergometer. This method enables measurement of maximal short-term power (P_max_) and corresponding optimal movement velocity (υ_opt_—velocity at which the power reaches a maximum value). Originally developed and validated in younger subjects, this methodology has been proved to provide accurate measurements of muscle function during brief non-isokinetic cycling, related to muscle fibre type composition [7,8]. Two repetitive tests with different loads have been proposed to obtain precise P_max_ and υ_opt_ measurements from the pooled data [8]. Consequently, the measurement technique was further developed and applied in populations of patients and older adults [9,10]. Nevertheless, the feasibility and reliability of this methodology has not been comprehensively assessed in different populations of patients and older subjects. Generally, the reliability of power measurements on a cycle ergometer was rather scarcely reported, except for in sportsmen and young adults [11]. Therefore, the goal of our study was to assess feasibility and reliability of quadriceps muscle power and contraction velocity measurements in different populations of subjects.

We aimed to investigate two aspects of reliability: (1) by comparing the repeatability of two measurements with different loads in four different populations of subjects, as previously demonstrated in young sportsmen by Arsac et al. [8]; (2) by comparing P_max_ and υ_opt_ calculated from the pooled data of the two measurements with P_max_ and υ_opt_ obtained with the second identically instrumented bicycle.

## 2. Materials and Methods

### 2.1. Participants

Five groups of community-dwelling subjects, fifty participants in each group, took part in the study. Participants were recruited during several consecutive years by the three University Departments: of Preventive Medicine, Rehabilitation, and Geriatrics. Respondents were recruited during several research and educational projects (see acknowledgments). An attempt was made to cover different healthy and clinical populations. The first group was fifty students (five males) of the Medical University. The second group was fifty male patients, participants of the ambulatory cardiac rehabilitation program, in whom the most recent acute coronary event, cardiac or cardio-surgery intervention had occurred a minimum of one month earlier. The third group was fifty patients (29 men) after stroke, in whom the most recent cerebrovascular event had occurred a minimum of one month earlier: 31 patients were 1–2 months, 15 patients were 2–3 months and 4 patients were more than 3 months after stroke. The fourth group was fifty older adults (aged ≥ 60 years; 14 men). The fifth group was fifty subjects of different ages (students, university employees and patients; 24 men) who volunteered to perform repetitive measurements with two different bicycles. All the participants had the ability to understand and execute commands with the ability to perform exercise testing, and signed the consent to participate in the study. Exclusion criteria were as follows: acute illness, unstable cardiovascular or metabolic disease, recent (<1 month) diagnosis of myocardial infarction, stroke or orthopaedic surgery, upper or lower limb amputation, cardiac contraindications to exercise tests, lack of ability to perform tests because of motor system dysfunctions (limited range of motion, pain, spasticity), and cognitive impairment. The proposal of this study was approved by the Bioethics Committee of the Medical University of Lodz, Poland (Ref. No RNN/338/08/KB). The study complies with the Declaration of Helsinki. All the participants signed the informed consent after hearing the explanation about the methods and aims of the study.

### 2.2. Protocol

Participants reported to the Research Center between 8 and 10 a.m. after overnight rest and a light breakfast, refraining from intense physical exercises, smoking and alcohol consumption for at least 12 h before measurements. Medical assessment was performed with each subject. All participants underwent medical examination, which included blood pressure measurement, to identify possible contraindications to the muscle power testing. During the interview, information (if lacking—already available in patients) on socio-economic status, current and previous illnesses and current medication was obtained. Body mass was also measured using the RADWAG personal weight scale (WPT60 150OW) (RADWAG Balances and Scales, Radom, Poland).

### 2.3. Maximum Power (P_max_) and Optimal Movement Velocity (υ_opt_) of the Knee Extensor Muscles

Mechanical measurements were performed on a friction-loaded cycle ergometer (Monark type 818E Stockholm, Sweden). The ergometer was instrumented with a strain gauge (KMM20 type, 200N, WObit, Poznań, Poland) and an incremental encoder (Rotapuls 141-H-200ZCU46L2 type, 200 pts/turn, Lika Electronic, Carre, Italy) for measurement of the friction force applied by the tension of the belt that surrounded the flywheel and the flywheel displacement, respectively. The strain gauge was calibrated with a known mass (4.0 kg). Flywheel inertia was calculated by the method suggested by Lakomy [12]. For all experiments, the saddle height was adjusted to give optimal comfort for each subject and toe clips were removed to prevent any pulling action of the contra lateral leg. Instantaneous pedaling velocity (υ), force (F) and power output (P) were calculated every 5 ms and then averaged over each downstroke period. The participants were first familiarized with the ergometer with 5 min of submaximal cycling and sprints of 3–4 s against different friction loads. After stretching and 5 min rest, the participants were asked to perform two 8 s sprints from a standardized starting position, each separated by at least 5 min rest. Friction loads were 0.25 N·kg^−1^ and 0.35 N·kg^−1^ of body mass in women and 0.25 N·kg^−1^ and 0.45 N·kg^−1^ of body mass in men, except in the case of the oldest (>80 years old) and the weakest subjects, in whom friction loads of 0.15 N/kg and 0.25 N/kg of body mass were used based on previous experience [10]. This allowed for at least 10–12 rotations during 8 s sprint and enabled function calculations from at least 20 obtained points. The υ-P combinations obtained during each sprint were fitted by a least square mathematical procedure to establish the υ-P relationship. The highest value of P (maximal short-term power—P_max_) and optimal movement velocity (υ_opt_—velocity at which the power reaches a maximum value) were calculated from a 3^rd^ order polynomial function [8]. P_max_ was expressed relative to body mass (W·kg^−1^). υ_opt_ was given in number of rotations per minute (rot·min^−1^). Measurements were performed with the program “Rowerki” developed by the Firma ZL, Zbigniew Lipinski (Lodz, Poland) and the Department of Geriatrics, Medical University of Lodz, Poland. For the four study groups (students, cardiac, stroke, older), the comparisons were performed between the two 8 s sprints. For the fifth group performing repetitive measurements with two different bicycles, after at least 15 min rest, the protocol of two 8 s sprints was repeated in a random order to obtain second measurements. For this group comparisons were made between the pooled data of the first bicycle and the pooled data of the second bicycle.

### 2.4. Statistical Analysis

Power calculation was performed to estimate the required sample size. The sample size required to detect differences of 10% in muscle function variables (with 0.80 power, 0.05 alpha type I error rate and given baseline standard deviation values) was 26 and 27 for P_max_·kg^−1^ and υ_opt_, respectively. Data were verified for normality of distribution (Shapiro–Wilk test) and equality of variances (Levene’s test) in each group. The results enabled the use of parametric tests. T-test for paired comparisons was used to compare mean differences between measurements (test1 vs. test2). Pearson’s correlation coefficients and Intraclass Correlations Coefficients (ICC) were calculated for repeated measurements. ICC (model 2,1) were interpreted as follows: values less than 0.5, between 0.5 and 0.75, between 0.75 and 0.9, and greater than 0.90 are indicative of poor, moderate, good, and excellent reliability, respectively. The standard error of measurement (SEM), representing the absolute reliability, was calculated as: SEM = standard deviation × √(1 − ICC). The coefficient of variation (CV or SEM%) was calculated as SEM divided by the mean and expressed in %. The minimal detectable change (MDC) (the minimum amount of change that can be interpreted as a real change) was calculated for the 95% confident interval (CI) as: MDC = SEM × 1.96 × √2. MDC% was calculated as MDC divided by the mean of the two tests and expressed in %. The agreement between the two measurements was graphically examined using the Bland–Altman approach: the difference between each pair of measurements was plotted against their mean. Heteroscedasticity was measured as correlation coefficients between mean differences and mean values of repetitive measurements. The results are presented as the mean ± standard deviation (SD). The level of significance was set at *p* < 0.05 for all the analyses. All statistical procedures were performed using the Statgraphics Plus 5 software package (Statpoint Technologies, Inc., The Plains, VA, USA) and Statistica 13.1 software package (StatSoft, Cracow, Poland).

## 3. Results

The baseline characteristics of the five study groups are shown in Table 1. The test–retest reliability of the P_max_ and υ_opt_ measurements in the five study groups is shown in Table 2. The correlations calculated for the pairs of scores ranged from 0.93 to 0.99 for P_max_ and from 0.86 to 0.96 for υ_opt_. All the correlations were highly significant (*p* < 0.001) and demonstrative of very good reliability. There was also good and excellent test–retest reliability measured by ICC (from 0.93 to 0.98 for P_max_ and from 0.86 to 0.95 for υ_opt_). All *t*-test values for paired comparisons were ≥ 0.05, showing no difference between repeated measurements. SEM varied from 16.9 to 21.4 W for P_max_ and from 2.91 to 5.54 rot/min for υ_opt_. The CVs for P_max_ and υ_opt_ in the stroke group were 10.6% and 11.4%, respectively; all other CVs were clearly lower that 10%. MDC varied from 46.6 to 59.3 W for P_max_ and from 8.07 to 15.4 rot/min for υ_opt_. MDC% varied from 9.53% to 29.3% for P_max_ and from 8.19% to 31.7% for υ_opt_, and was the highest in the stroke group. 

The Bland–Altman plots (Figure 1) graphically show the reliability patterns in terms of systematic errors and the limits of agreement between the repeated measures in older adults and tests performed with the two bicycles. The heteroscedasticity analysis showed no association between inter-trial mean differences and inter-trial averages (all the associations were statistically not significant) (Table 2). 

Example of power–velocity curves of tests performed with two bicycles is shown in Figure 2.

## 4. Discussion

In this study, we have shown that muscle power and muscle contraction velocity measurements during brief maximal exertion on a friction-loaded cycle ergometer are feasible in older subjects and those with diseases. All the tests and indices of relative reliability indicated very good repeatability in all the studied groups. Likewise, the SEMs, CVs, and MDCs as the measures of absolute reliability in all except the stroke group showed very low indices of error. In the stroke group, CVs were more than 10% but still at an acceptable level.

### 4.1. Muscle Strength and Functional Tests

Previous reliability studies concentrated mainly on muscle strength or functional tests assessment. In independently living older subjects, the ICCs varied from 0.81 to 0.99 for the inter- and intra-rater reliability of isokinetic strength of the knee and ankle [13]. The ICC of the test–retest ranged from 0.60 for the ankle extensors to 0.85 for the elbow flexors and 0.87 for the elbow extensors with a hand-held dynamometer in thirty nursing home residents. The MDC% varied from 24.38 (elbow extensors) to 81.97 (ankle extensors) [14]. As summarized in a systematic review, the ICCs of handgrip strength dynamometry relative test–retest reliability were at least 0.80 in the majority of screened studies. Nevertheless, the ICCs were lower in some studies with older adults with mental illnesses, and the MDC% ranged from 14.5% to 98.5% [15].

Functional tests, usually requiring some strength and power, were widely assessed in older populations and those with diseases. The ICCs reported earlier for the five-repetition sit-to-stand test in the article ranged from 0.64 to 0.96 [16]. As recently reviewed, the test–retest reliability of the five-repetition sit-to-stand test in adults was high (ICC = 0.937) [17]. Nevertheless, it is often not feasible in frail subjects with a lower level of functioning, and some modifications have been proposed [18]. For the Modified Figure of Eight, the ICC ranged from 0.73 to 1.0 [19]. Reproducibility (CVs) varied from good (<10%) to moderate (17.2–18.2%) for a counter-movement jump in moderately trained older individuals [20]. Two large Canadian studies revealed that in reality, the percentage of older adults not able to perform some functional tests is quite high, and that confirmative measurements may have a lower performance as compared to the original ones. In the first study, 29.3% of older subjects were not able to perform the Timed Up and Go (TUG) test, and 35.9% were incapable of completing the Functional Reach test, mainly because of dementia. The ICCs between the first and the second administrations of the TUG ranged from 0.50 to 0.56 [21]. In the second study, the ICCs of relative reliability varied from 0.64 for the chair rise test and gait speed, to 0.78–0.82 for the single-leg stance test and TUG, and 0.95 for handgrip strength [22].

### 4.2. Power Measurements

The reliability of different power measures in tests of physical performance has been assessed in meta-analysis. Field and laboratory tests of sprint running had the smallest CV (approximately 0.9%), power tests lasting 1 min to 3 h on a treadmill or cycle ergometer had a CV of 0.9 to 2.0%, lactate-threshold power tests had a CV of approximately 1.5%, and height or distance jump had CVs of approximately 2.0%. The CV for the mean power on isokinetic ergometers was >4% [23]. A dynamometer for the evaluation of dynamic muscle work had correlation coefficients of 0.88 to 0.97 [24]. Inter-rater reliability (ICC) measured with Biodex was 0.85 for the estimated unloaded velocity and 0.99 for P_max_ [3].

As reviewed by Driss and Vandewalle, in young adults, the reliability of the results of the cycling Wingate test measured by the test–retest coefficient of correlation is good for the peak and mean power (r > 90) [11]. In physical education students, the ergometer values of r (test–retest) and ICC were higher than 0.9, and those of SEE were lower than 5% for P_max_ [25]. In another study with physical education students, cycling peak power test–retest CVs were only about 3% [26]. Therefore, the reliability of ergometer-measured peak power in younger and healthy adults is generally high, independent of the protocol [11].

### 4.3. Power Measurements in Older Subjects and Those with Diseases

In the reference study measuring the explosive power of the leg with the Nottingham power rig in 46 subjects aged 20 to 86 years, the CV was 9.4% [27]. The CV observed by the examiners was 3.46–3.49% with the Nottingham power rig in 55 men aged 73 yrs on average [28]. In 72 adults aged 63 years, the ICCs were 0.88–0.96 [29].

In older women who underwent lower-extremity muscle power measurements with Biodex, the ICCs were 0.90–0.97 for knee extension, 0.83–0.96 for knee flexion, and 0.83–0.96 for plantar flexion and dorsiflexion power. The CVs varied from 9.9 to 20.0% [30]. In older men, Pearson correlation coefficients ranged from 0.74 to 0.96, and the CVs were generally less than 10%, but for leg extension power, the CV was 15.5% [31]. The ICC for the isometric rate of torque development was low (0.44), while that of isokinetic leg extension power was moderate to high (0.80 and 0.91 for two different isokinetic velocities) in older men. The CVs for muscle power were 6.19% and 16.1% [32]. The ICC for dorsiflexion and plantar flexion power measures assessed after seven days in older women varied from 0.86 to 0.97 [33].

For a 9-second modified Wingate Anaerobic Test, the ICC of the first and second values of mean power was 0.982 in 28 hemiplegic stroke patients [34]. The ICC was 0.87–0.98 for isotonic quadriceps power measures, and the SEM was < 10% for average/peak power and peak velocity in patients with mild to very severe COPD [35].

The force–velocity tests used in the present study allowed for the measurement of the force and velocity components of power. Multiple υ-P combinations allow us to establish the precise υ-P relationship and calculate the highest value of P (P_max_) and the corresponding optimal movement velocity (υ_opt_—velocity at which the power reaches a maximum value) [8]. The age-related decline in Ʋ_opt_ is steeper than that in muscle strength. In some studies, Ʋ_opt_ has been suggested to be the better marker of muscle functional decline as compared to muscle strength [5,36]. The precision of measurements of P_max_ and υ_opt_ in our study was confirmed by very good indices of absolute and relative reliability.

The feasibility of many functional tests is restricted in numerous patients, e.g., after stroke. With the present measurement, only subjects with severe disability were not able to perform the test. Our many years of experience indicate that the method is safe; no complications or side-effects were recorded. The method is not time-consuming and relatively not expensive. All of those features makes the friction-loaded cycle ergometer instrumented with a strain gauge and an incremental encoder an excellent candidate for future clinical studies in many older and populations and those with diseases, e.g., in patients after stroke or a cardiac event. Based on previous experience, we recommend friction loads 0.25 N·kg^−1^ and 0.35 N·kg^−1^ of body mass in women and 0.25 N·kg^−1^ and 0.45 N·kg^−1^ of body mass in men, except in the case of the oldest (>80 years old) and the weakest subjects, in whom friction loads of 0.15 N/kg and 0.25 N/kg of body mass should be used. This allows for at least 10–12 rotations during 8 s sprints and allows for function calculations from at least 20 obtained points. 

### 4.4. Limitations

Several shortcomings of the present study should be acknowledged. We tested different non-homogenous populations with over-representations of women or men, depending on the group. Subjects with severe dementia, frailty, and disability were not able to perform the test, leading to some selection bias. In stroke patients, the reliability parameters were lower but still at an acceptable level. In the present study, toe clips were removed to prevent any pulling action of the contra lateral leg. For the hemiplegic stroke patients with the most severe disabilities, toe clip removal may be reconsidered in order to allow them to perform the test. This should be the subject of future separate study(ies). Validation with various functional tests in different clinical populations should also be targeted in future studies.

## 5. Conclusions

The proposed methodology is precise and feasible in older subjects and those with diseases, except for those with a severe disability. Therefore, the friction-loaded cycle ergometer may be a worthwhile candidate for future clinical studies, and the proposed methodology may permit the monitoring of functional status during rehabilitation in those populations.

## Figures and Tables

**Figure 1 biology-13-00140-f001:**
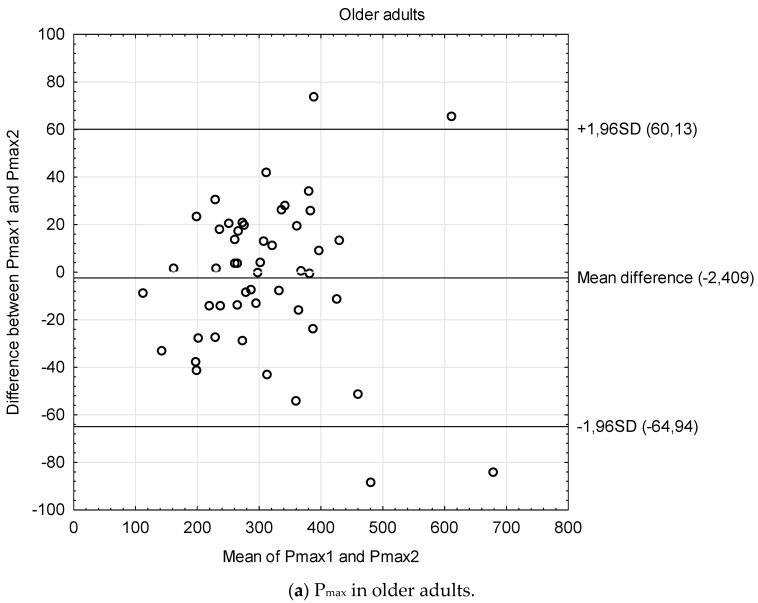
The Bland-Altman plots showing the reliability patterns in terms of systematic errors and the limits of agreement between the repeated measures.

**Figure 2 biology-13-00140-f002:**
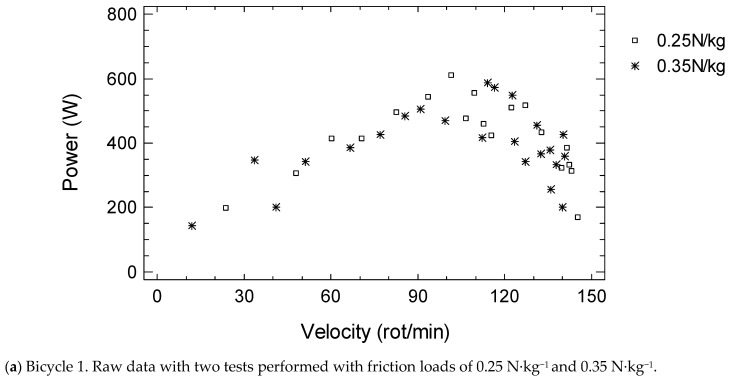
Example of power-velocity curves of tests performed with two bicycles in 37-year old women.

**Table 1 biology-13-00140-t001:** Baseline characteristics of the five study groups.

	Students	Cardiac Patients	Stroke Patients	Older Adults	Bicycle 1 vs. 2
Age (years)	24.1 ± 1.1	54.9 ± 7.58	66.7 ± 8.6	72.6 ± 5.6	47.6 ± 19.8
Body mass (kg)	60.1 ± 10.4 W-57.5 ± 6.1 M-84.2 ± 10.5	82.6 ± 8.9	76.9 ± 14.7 W-70.6 ± 14.2 M-81.4 ± 13.5	72.4 ± 12.4 W-70.6 ± 12.7 M-76.9 ± 10.6	68.1 ± 10.9 W-59.5 ± 6.9 M-77.3 ± 8.7
Sex [number (%)]	W-45(90%) M-5(10%)	W-0 M-50 (100%)	W-21(42%) M-29(58%)	W-36(72%) M-14(28%)	W-26 (52%) M-24 (48%)

W—women; M—men.

**Table 2 biology-13-00140-t002:** Test–retest reliability of P_max_ and υ_opt_ measurements in the five study groups.

Group		Students		Cardiac Patients		Stroke Patients		Older Adults		Bicycle 1 vs. 2	
N		50		50		50		50		50	
Measurement		P_max_ (W)	υ_opt_ (rot/min)	P_max_ (W)	υ_opt_ (rot/min)	P_max_ (W)	υ_opt_ (rot/min)	P_max_ (W)	υ_opt_ (rot/min)	P_max_ (W)	υ_opt_ (rot/min)
Test 1	$\overset{¯}{x}$	460.0	101.7	570.5	98.8	157.2	48.8	310.5	74.8	417.6	89.4
	SD	139.7	10.5	113.7	13.5	60.7	15.6	106.2	12.4	134.9	19.2
Test 2	$\overset{¯}{x}$	454.2	100.9	569.4	98.2	160.8	48.1	312.9	74.1	415.3	88.2
	SD	135.7	11.2	112.1	12.5	66.4	14.1	108.3	13.4	134.0	18.6
Test 1-test 2 (r)		0.98	0.93	0.97	0.95	0.93	0.86	0.96	0.88	0.99	0.96
*t*-test		1.35	−1.19	0.29	1.0	−1.05	0.57	−0.53	0.75	0.83	1.45
P for *t*-test		0.18	0.24	0.77	0.30	0.30	0.57	0.60	0.46	0.48	0.15
ICC		0.98	0.92	0.97	0.95	0.93	0.86	0.96	0.87	0.98	0.94
SEM		19.5	3.08	19.6	2.91	16.9	5.54	21.4	4.65	19.0	4.63
CV (SEM%)		4.27	3.04	3.44	2.95	10.6	11.4	6.87	6.24	4.56	5.21
MDC		54.1	8.54	54.3	8.07	46.6	15.4	59.3	12.9	52.7	12.8
MDC%		11.8	8.43	9.53	8.19	29.3	31.7	19.1	17.3	12.7	14.4
Heterosc edasticity	R	−0.13	0.17	−0.06	−0.24	−0.24	0.20	0.07	0.15	0.027	−0.006
	*P*	0.36	0.24	0.66	0.09	0.09	0.16	0.64	0.30	0.85	0.97

ICC—intraclass correlation coefficient, SEM—standard error of measurement, CV—coefficient of variation, MDC—minimal detectable change at a 95% confidence level.

## Data Availability

Data used in the present study will be made available upon reasonable request to the corresponding author.

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
