# Peer review of "Feasibility and Reliability of Quadriceps Muscle Power and Optimal Movement Velocity Measurements in Different Populations of Subjects"

_biology, 2024, doi:10.3390/biology13030140_

Round 1

Reviewer 1 Report

Comments and Suggestions for Authors

The paper is well-written with minor corrections required to the text. The methods used to assess the feasibility and reliability of muscle power and movement velocity using a friction loaded cycle ergometer seem appropriate, but do not validate the measurement method. This may be a drawback in its clinical application in the targeted population(s).

A rationale for the selecting groups should be provided, as the justification for old and diseased subjects seems vague. How old? Which diseases?

It is mentioned that the Shapiro-Wilk test of normality  was used, but no information is provided about the statistical inference processes requiring such assumption, nor the results of that normality test. Was is for the t test? In that case, was it applied to Test 1 and Test 2 data in separate, or to the difference? How was the bivariate normality assumption required by testing the Pearson correlation coefficient addressed by the authors?

Still in the statistics field, the authors write (p. 4, ll.  168-169): "All t-test 168 values for paired comparisons were less than 0.05, showing no difference between repeated measurements.". Unless the significance level used by the authors is lower than 5%, which is not the case, in view of what is stated between ll. 157-160, this statement seems incorrect. Please clarify.

Finally, what is the robustness of the mathematical models described in association with figure 2,. given that is refers to the performance of a single subject? 

To conclude, before the paper is accepted, the authors should address these questions adequately, as they are relevant for the validity of their conclusions.

Comments on the Quality of English Language

Some minor errors in the text must be eliminated by the authors.

Author Response

Answers to the Reviewer 1

We thank the Reviewer for all the constructive comments.

The paper is well-written with minor corrections required to the text. The methods used to assess the feasibility and reliability of muscle power and movement velocity using a friction loaded cycle ergometer seem appropriate, but do not validate the measurement method. This may be a drawback in its clinical application in the targeted population(s).

Answer: We do agree that the validation with different procedures assessing muscle function would be appropriate. This was done with young athletes and was cited in the current study. Validation in different clinical populations will be targeted in our future studies. The need for such a validation has been added in the final part of the discussion.

A rationale for the selecting groups should be provided, as the justification for old and diseased subjects seems vague. How old? Which diseases?

Answer: A rationale for the selecting groups has been provided as suggested. We tried to cover different populations. Nevertheless, the need for other clinical settings has been mentioned according to the suggestion of the Reviewer.

It is mentioned that the Shapiro-Wilk test of normality  was used, but no information is provided about the statistical inference processes requiring such assumption, nor the results of that normality test. Was is for the t test? In that case, was it applied to Test 1 and Test 2 data in separate, or to the difference? How was the bivariate normality assumption required by testing the Pearson correlation coefficient addressed by the authors?

Answer: Pmax and Vopt data were normally distributes with equal variances for repeated measurements. T-test for paired comparisons was used, so normality assumptions were assessed separately for test1 and test2 data and equality of variances was measured between test1 and test2 data. This enabled using Pearson correlations for repeated measurements. All this information has been presented more clearly in the Statistical analysis section.

Still in the statistics field, the authors write (p. 4, ll.  168-169): "All t-test 168 values for paired comparisons were less than 0.05, showing no difference between repeated measurements.". Unless the significance level used by the authors is lower than 5%, which is not the case, in view of what is stated between ll. 157-160, this statement seems incorrect. Please clarify.

Answer: Thank for pointing to this obvious linguistic error. All p-values for paired comparisons were ≥0.05 (NS) as shown in the Table 2. This mistake has been corrected in the text and exact p-values have been provided in the Table 2. 

Finally, what is the robustness of the mathematical models described in association with figure 2,. given that is refers to the performance of a single subject? 

Answer: The robustness of the mathematical models has been given:

R squared =79.7; R squared (adjusted for d.f.) =78.3.

R squared =78.7; R squared (adjusted for d.f.) =77.2.

To conclude, before the paper is accepted, the authors should address these questions adequately, as they are relevant for the validity of their conclusions.

Answer: Thank you again. All the comments have been introduced to the manuscript.

Some minor errors in the text must be eliminated by the authors.

Answer: The manuscript has been reviewed by an English Editor.

Reviewer 2 Report

Comments and Suggestions for Authors

The authors use five groups to validate a measurement system, they compare, in four groups, two different situations (friction loads) and in another group two identical ergometers. They provide data on the reliability of the tests. We are not surprised by the results obtained; they are very predictable.

1.       The authors do not indicate how they measured body mass. Nor do they say with what instrument.

2.       The paragraph in lines 102-106 is not a method. It doesn't need to be included.

3.       In section 2.1 Participants, in the group of stroke patients, the authors do not describe the presence of sequelae (hemiplegics? paresis?).

4.       Line 121 the authors write "... most disabled subjects..." I don't understand which group these disabled people belong to, with the exclusion criteria they shouldn't be part of the study.

5.       In line 123 the authors write: "...enables function calculations from at least 20 obtained points". I don't understand the 20 points. Nowhere else in the work is there talk of points.

6.       The authors do not indicate how they calculated the sample size.

7.       They use the S-W test to see the normality of the population distribution. For the number of subjects, the Kolmogórov-Smirnov  (K-S)  test is more appropriate.

8.       In Table 1, authors should write the units of measurement for each variable and decipher the abbreviations at the bottom of the table.

9.       Table 2 would be better if you replaced the abbreviation NS with its real value in the "P for t-test" row. Just like they did in row P – Heteroscedasticity.

 10.   Sections 4.2; 4.2 and 4.3 of the discussion are very well documented, there is a lot of data extracted from the original articles, but the authors only cite them. There is no clear and explicit relationship with the authors' data, methodology or ideas. In other words, I think there is a lack of discussion. Authors need to improve on this aspect.

 11.   This is a basic research work, I think that in the discussion there is a section missing, or a few paragraphs, dedicated to the possible clinical or practical application of its findings. This could be done, both globally and separately for each of the study group populations.

 12.   What is the interest and what does this test bring to the cycle ergometer in stroke patients?

13.   In a practical application of this test that Friction loads recommend the authors and why?

 14.   On limitations, the authors write: "...toe clips were removed to prevent any pulling action of the contra lateral leg". This is not described in the method and contradicts the exclusion criterion: "...lack of ability to perform tests because of motor system dysfunctions (limited range of motion, pain, spasticity)...". Authors should clarify these aspects and unify criteria.

 15.   The authors should remove the following sentence from the conclusions: "Our many-year experience indicates that the method is safe – no complications or side-effects were recorded. Finally, the method is not time consuming and relatively not expensive". The objectives of this work are not to analyze the complications, nor is it an objective to determine or assess the time spent, nor the cost of the tests. These statements can be included in the discussion by completing a section on practical or clinical applications.

Author Response

Answers to the Reviewer 2

We thank the Reviewer for all the constructive comments.

The authors use five groups to validate a measurement system, they compare, in four groups, two different situations (friction loads) and in another group two identical ergometers. They provide data on the reliability of the tests. We are not surprised by the results obtained; they are very predictable.

Answer: On one hand, the results may be considered as predictable, but this is due, in our opinion, to very precise physiological measurements applied. On the other hand, reliability data seem rather better as compared to the other power measurements methods. This again is the result of very precise assessment that stems from drawing multiple measurement points.

  1. The authors do not indicate how they measured body mass. Nor do they say with what instrument.

Answer: Patients were weighed on RADWAG personal weight scales (WPT60 150OW) (RADWAG Balances and Scales, Radom, Poland). This information has been provided.

  1. The paragraph in lines 102-106 is not a method. It doesn't need to be included.

Answer: This paragraph was removed from Methods as suggested.

  1. In section 2.1 Participants, in the group of stroke patients, the authors do not describe the presence of sequelae (hemiplegics? paresis?).

Answer: 31 patients were 1-2 months, 15 patients were 2-3 months and 4 patients were more than 3 months after stroke. 39 patients were after the first, 11 patients were after the second or third stroke. 5 patients had haemorrhagic and 45 ischemic stroke; 26 left, 24 right side stroke. The consequences of stroke could not be debilitating to enable exercise testing. All the participants had ability to understand and execute commands with the ability to perform exercise testing. Exclusion criteria included recent (<1 month) diagnosis of stroke or orthopaedic surgery, upper or lower limb amputation, lack of ability to perform tests because of motor system dysfunctions (limited range of motion, pain, spasticity), and cognitive impairment. This information has been provided.

  1. Line 121 the authors write "... most disabled subjects..." I don't understand which group these disabled people belong to, with the exclusion criteria they shouldn't be part of the study.

Answer: "... most disabled subjects..." has been changed to "... the weakest subjects..."

  1. In line 123 the authors write: "...enables function calculations from at least 20 obtained points". I don't understand the 20 points. Nowhere else in the work is there talk of points.

Answer: Graphical examples of calculations have been presented in Figures 2a-2d.

  1. The authors do not indicate how they calculated the sample size.

Answer: Sample size calculation has been presented in Statistical methods, lines 138-141.

  1. They use the S-W test to see the normality of the population distribution. For the number of subjects, the Kolmogórov-Smirnov(K-S)  test is more appropriate.

Answer: Both tests are corrects. They give comparable results.

  1. In Table 1, authors should write the units of measurement for each variable and decipher the abbreviations at the bottom of the table.

Answer: Units of measurement have been presented. Abbreviations have been explained.

  1. Table 2 would be better if you replaced the abbreviation NS with its real value in the "P for t-test" row. Just like they did in row P – Heteroscedasticity.

Answer: Exact p-values have been provided.

  1. Sections 4.2; 4.2 and 4.3 of the discussion are very well documented, there is a lot of data extracted from the original articles, but the authors only cite them. There is no clear and explicit relationship with the authors' data, methodology or ideas. In other words, I think there is a lack of discussion. Authors need to improve on this aspect.

Answer: We do agree that there is quite a lot of data on reliability of different muscle strength and functional measurements in different populations. We tried to summarized those data especially concentrating on available power and velocity measurements. The comparison of present results with previous data gives some picture on the reliability in different approaches. Nevertheless, according to the suggestion of the Reviewer, the discussion has been modified to present more clearly the relationship of previous studies with regard to our data.

  1. This is a basic research work, I think that in the discussion there is a section missing, or a few paragraphs, dedicated to the possible clinical or practical application of its findings. This could be done, both globally and separately for each of the study group populations.

Answer: The paragraph has been added discussing the possible clinical and practical application of its findings, also both globally and separately for each of the study group populations.

  1. What is the interest and what does this test bring to the cycle ergometer in stroke patients?

Answer: Stroke patients are one of the most vulnerable groups. Feasibility of many functional tests is restricted in this group of patients. Proposed in the present study methodology enables precise measurements in many of those patients, and this permits monitoring of functional status during rehabilitation. These considerations have been presented more clearly in the discussion.

  1. In a practical application of this test that Friction loads recommend the authors and why?

Based on previous experience we recommend friction loads 0.25 N·kg−1 and 0.35 N·kg−1 of body mass in women, 0.25 N·kg−1 and 0.45 N·kg−1 of body mass in men, except in the case of the oldest (>80 years old) and the weakest subjects, in whom friction loads of 0.15 N/kg and 0.25 N/kg of body mass should be used. This allows at least 10-12 rotations during 8 s sprint and enables function calculations from at least 20 obtained points.

These information has been presented more clearly in the methods, results and discussion.

  1. On limitations, the authors write: "...toe clips were removed to prevent any pulling action of the contra lateral leg". This is not described in the method and contradicts the exclusion criterion: "...lack of ability to perform tests because of motor system dysfunctions (limited range of motion, pain, spasticity)...". Authors should clarify these aspects and unify criteria.

Answer: It has been stated in the methods that:

“For all experiments the saddle height was adjusted to give optimal comfort for each subject and toe clips were removed to prevent any pulling action of the contra lateral leg.”

This was applied for all studied groups

  1. The authors should remove the following sentence from the conclusions: "Our many-year experience indicates that the method is safe – no complications or side-effects were recorded. Finally, the method is not time consuming and relatively not expensive". The objectives of this work are not to analyze the complications, nor is it an objective to determine or assess the time spent, nor the cost of the tests. These statements can be included in the discussion by completing a section on practical or clinical applications.

Answer: According to the suggestion of the Reviewer, above statements have been removed from the conclusions and included in the discussion.

Round 2

Reviewer 2 Report

Comments and Suggestions for Authors

Regarding my comment number 7: Indeed, both the Shapiro Wilk test (S-W) and the Kolmorov Smirnov test (K-S) are valid for determining the normal distribution of the population. Statisticians indicate that the K-S is more suitable for populations of more than 50 subjects and the S-W for populations with fewer subjects. If, to determine the normal distribution of variables, you have studied the entire population (250 subjects), the correct thing to do would be to perform the K-S test. If you have studied it in each group independently, having 50 subjects each, from my point of view, you could use any of them. But you must specify it.

Author Response

Author’s reply to the Reviewer’s 2 comments (Round 2)

Regarding my comment number 7: Indeed, both the Shapiro Wilk test (S-W) and the Kolmorov Smirnov test (K-S) are valid for determining the normal distribution of the population. Statisticians indicate that the K-S is more suitable for populations of more than 50 subjects and the S-W for populations with fewer subjects. If, to determine the normal distribution of variables, you have studied the entire population (250 subjects), the correct thing to do would be to perform the K-S test. If you have studied it in each group independently, having 50 subjects each, from my point of view, you could use any of them. But you must specify it.

Answer: Thank you for pointing to this important issue. Indeed, we have studied the normality of distribution in each group independently, having 50 subjects each. This has been clearly acknowledged according to the suggestion of the Reviewer.